# Evaluation of Medication Adherence and Appropriateness Among Heart Failure Patients Attending the Cardiac Clinic at a Tertiary Care Hospital: A Cross-Sectional Observational Study

**DOI:** 10.3390/pharmacy13040101

**Published:** 2025-07-27

**Authors:** Nanayakkara Muhandiramalaya Yasa Kalum Bagyawantha, Isuri Nilnuwani Dangahage, Ghanamoorthy Mayurathan, Weerasinghe Mudiyanselage Suminda Pushpika

**Affiliations:** 1Department of Pharmacy, Faculty of Allied Health Sciences, University of Peradeniya, Peradeniya 20400, Central Province, Sri Lanka; isurinilnuwani@gmail.com; 2National Hospital Kandy, Kandy 20000, Sri Lanka; gkmayurathan@yahoo.com (G.M.); sumindapushpika@gmail.com (W.M.S.P.)

**Keywords:** heart failure, medication adherence, medication appropriateness

## Abstract

Heart failure is a chronic disease with significantly high morbidity and mortality rates, and a thorough understanding of medication adherence and appropriateness is crucial to ensure effective treatment outcomes. This cross-sectional observational study aimed to assess medication adherence, understand the influence of sociodemographic factors on medication adherence, and assess the medication appropriateness for heart failure patients attending the cardiac clinic at National Hospital Kandy (NHK). This study was conducted among 325 heart failure patients attending the cardiac clinic at the NHK. Medication adherence was assessed using the brief medication questionnaire (BMQ) after detailed medication history interviews. Statistically significant associations between total BMQ scores and sociodemographic factors were determined at 95% confidence interval. The appropriateness of the newly prescribed medication lists was assessed using the medication appropriateness index (MAI). Among the 325 patients recruited, the mean total BMQ score was 1.16; 11.7% of the participants were adherent to their medications whereas 15.4% had poor adherence. Most participants (52.0%) were in the probable poor adherent level. Statistically significant associations were observed between total BMQ score and age, sex, and education level. The mean MAI score was 0.56. Medication adherence among heart failure patients was poor and some sociodemographic factors influenced medication adherence. The appropriateness of prescribed medications was found to be acceptable.

## 1. Introduction

### 1.1. Cardiovascular Diseases and Heart Failure

Cardiovascular diseases (CVDs) are clinical conditions that impair the proper function of the heart or blood vessels in the heart [1]. Currently, CVDs are known as the leading cause of morbidity and mortality worldwide, resulting in approximately 17.9 million deaths each year [2,3,4]. Coronary heart disease, stroke, and congestive heart failure are considered the common types of CVD and other forms include congenital cardiovascular disorders, atrial and ventricular arrhythmias, and rheumatic heart disease [4].

Heart failure is considered a notable global public health burden and the total global prevalence of heart failure cases increased by 106.3% from 1990 to 2019 [5]. Heart failure is caused by any structural or functional cardiac abnormalities leading to impaired ventricular filling or ejection of blood, resulting in reduced cardiac output and/or increased intracardiac pressure [3,6]. The left ventricular ejection fraction (LVEF) is clinically used as a phenotype marker to categorise heart failure patients [7]. The prevalence of heart failure is estimated to increase by approximately 46% from 2012 to 2030, leading to an increase in global healthcare costs of approximately 127% [8]. Additionally, the prevalence of heart failure is projected to be approximately 64 million people globally, and the number of patients who live with heart failure has been increasing because of the ageing population, the growth of the global population, and improved survival after the diagnosis of the disease [7,8].

### 1.2. Medication Adherence and Importance

According to the World Health Organisation (WHO), medication adherence can be defined as “the extent to which a person’s behaviour corresponds with agreed recommendations from a health care professional” [9]. According to the ESPACOMP Medication Adherence Reporting Guidelines (EMERGE), the definition of medication adherence contains three phases, known as the initiation phase, implementation phase, and discontinuation phase, based on how long a patient has used a medication [10]. Initiation refers to when a patient takes the first dose of a prescribed medication. Implementation phase refers to taking the actual dosing regimen from first dose to when the last dose is taken. Discontinuation occurs when the patient stops taking the medication on their own, without medical advice [10]. CVDs were the leading cause of death from non-communicable diseases in the world, accounting for at least 19 million deaths in 2021 [11]. According to the WHO, only about 50% of patients with chronic diseases in developed countries adhere to their medications, with adherence rates being even lower in developing countries [12]. However, the rate of medication adherence among patients with chronic diseases is lower than that among patients with acute diseases [13]. Furthermore, long-term medication therapy is needed for the effective management of heart failure. It is a critical process and proper management enhances patient’s quality of life, and reduces risk of hospitalization and mortality [14]. Inadequate medication adherence has led to healthcare resource waste, exacerbation of the conditions, increased rate of hospitalisation, and decreased quality of life in patients [13,14,15]. Previous studies have shown that nonadherence rate among heart failure patients can range from 23 to 47% [16,17].

Factors affecting medication nonadherence among heart failure patients can be classified into patient-related, disease-related, regimen-related, healthcare provider-related, and socioeconomic-related factors [18]. Hence, medication adherence among heart failure patients can be affected by sociodemographic factors such as age, sex, marital status, and educational level [16,19]. Therefore, understanding and addressing the factors influencing medication adherence is essential to develop effective interventions aimed at improving therapeutic outcomes among heart failure patients [20].

### 1.3. Medication Appropriateness

Medication appropriateness can be defined as the evaluation of medical regimens and the identification of potentially inappropriate medicines (PIMs) [21]. Polypharmacy, also known as the use of multiple medications, is associated with an increased possibility for the occurrence of PIM in medication regimens [21,22]. Hence, assessing the medication appropriateness of medication regimens is necessary for the identification of drug–drug interactions, drug–disease interactions, and any therapeutic duplications. It is also an essential approach to ensure the effectiveness of treatment and minimise possible risks [21,23]. Most patients with heart failure suffer from various noncardiac comorbidities, such as diabetes, anaemia, and chronic pulmonary disease, resulting in an increased risk of medication inappropriateness [22]. More importantly, the European Society of Cardiology guidelines for heart failure management recommend reducing polypharmacy by stopping medications that do not affect symptom relief, patient prognosis, or quality of life [24]. Since most of the drug-related problems leading to medication inappropriateness are considered preventable issues, assessing medication appropriateness can be of pivotal importance in improving patient care in heart failure patients [25].

### 1.4. Assessing Medication Adherence and Appropriateness

In managing chronic diseases such as heart failure, assessment of both medication adherence and appropriateness is vital to ensure optimal outcomes among patients. Poor medication adherence and inappropriately prescribed medications are associated with disease progression, increased hospital readmissions, and increased mortality rates [26,27,28]. Medication adherence and improved patient outcomes are influenced by the collaborative efforts of many healthcare professionals. Within this multidisciplinary framework, each professional contributes uniquely to patient care. In this study, a pharmacy undergraduate under the supervision of a pharmacist was engaged in assessing medication adherence and appropriateness among the heart failure patients.

Heart failure is a condition that requires complex, appropriate long-term therapy and proper adherence to prescribed medications to achieve optimal outcomes of patients. Due to the limited availability of local data on medication adherence levels among heart failure patients, this study was conducted to explore medication adherence levels, evaluate the appropriateness of prescribed medications, and determine whether any associations exist between sociodemographic factors and medication adherence among heart failure patients attending the cardiac clinic at the NHK.

## 2. Materials and Methods

### 2.1. Study Design and Setting

The present study was a cross-sectional observational study conducted at the cardiovascular clinic at the NHK, Sri Lanka.

### 2.2. Ethical Considerations

Ethical approval for the study was obtained from the Ethics Review Committee of the Faculty of Allied Health Sciences, University of Peradeniya, Sri Lanka (AHS/ERC/2024/026), on 25 April 2024. Permission for the study was obtained from the director of the NHK and the staff in charge of the cardiac clinic before data collection. Participants were recruited for the study after written and verbal informed consent was obtained. The collected data were stored in a university-managed Google Workspace (G-Suite) cloud storage system, secured with password protection and two-factor authentication linked to institutional email accounts. Access to the raw data was strictly limited to the first and second authors, both of whom are affiliated with the university and directly involved in data handling and analysis. Only de-identified, analysed data were shared with other co-authors for interpretation and manuscript preparation. The study adhered to institutional data governance policies, and all procedures were carried out in accordance with applicable data protection regulations. The confidentiality and privacy of all participants’ information were strictly maintained throughout the research process.

### 2.3. Study Participants

Patients who were diagnosed with heart failure attending the cardiac clinic at NHK were selected for this study. Patients’ diagnoses mentioned in the clinic books and echocardiogram (ECHO) reports were used as the basis for recruiting patients to the study. Patients who were older than 18 years were included in this study, and patients who were pregnant, diagnosed with psychiatric disorders, admitted to the wards while attending the clinic, or had communication difficulties were excluded.

### 2.4. Sample Size

A convenient sampling method was used in this study. The required sample size was 385, which was calculated on the basis of a single population proportion formula in a cross-sectional survey [29].

### 2.5. Recruitment of Participants

The selected patients were informed about the study objectives and other relevant information with a written information leaflet. The patients were subsequently recruited and written, and verbal consent was obtained.

### 2.6. Study Tools

Sociodemographic data were obtained via a self-administered questionnaire, whereas the brief medication questionnaire (BMQ) and the medication appropriateness index (MAI) were used as interviewer-administered study tools to assess medication adherence and clinic medication appropriateness, respectively. The BMQ is a freely available, validated tool used to assess medication adherence, and it was translated into the local language of Sri Lanka in a previous study. The translated BMQ has demonstrated significant reliability, temporal stability, and validity [30]. Although BMQ is typically designed as a self-administered tool, it was used as an interviewer-administered tool because of the limited health literacy among many patients, and difficulty in recalling or pronouncing drug names. Patients’ clinic books were used as a reference during the interview process to ensure data accuracy. Other studies have also employed the BMQ as an interviewer- administered tool in similar healthcare settings [31]. The BMQ consists of three major screens: the regimen, belief, and recall screens. BMQ total scores of zero or close to zero were considered high medication adherence. Furthermore, patients can be divided into adherent, probable adherent, probable poor adherent, and poor adherent categories on the basis of the score they obtain for each sub screen in the BMQ tool [32]. There are sets of questions on each screen of the BMQ, and scoring is performed for each question depending on the patient’s answer. The correct answers were scored as zero and incorrect answers were scored as one. The total BMQ score was subsequently calculated, and zero was considered optimal medication adherence, whereas any positive scores were considered suboptimal medication adherence [33]. In the MAI, a score of zero per patient indicates a completely appropriate medication list on the basis of the patient’s condition, and when the score is high, appropriateness is considered low [34]. In this study, the criterion concerning the cost compared with equal alternatives in the MAI tool was not considered for the assessment because patients attending the cardiovascular clinic at the NHK were given their medications free of charge from the hospital.

### 2.7. Study Procedure

The study was carried out by a final-year undergraduate pharmacy student after obtaining the necessary training from clinical pharmacy educators under the observation of a practising pharmacist. All the activities carried out by the researcher were evaluated by expert clinical pharmacy educators. The medication history was recorded from each recruited patient to list their previous month’s clinic medication list to assess their medication adherence using BMQ. The study focused on individuals who were actively taking medications, indicating that they were in the implementation phase of medication adherence [10]. Patients were categorised into four groups based on medication adherence. Age, sex, marital status, education level, living arrangement, and employment status were collected, and the impact of those parameters on medication adherence was evaluated. A new medication plan for each patient after visiting the doctor was recorded, and the appropriateness of those medications was assessed using the MAI. MAI assessment of the new medication plan was carried out on the basis of the European Society of Cardiology Guidelines [35], the Australian Medicines Handbook (AMH), and the British National Formulary (BNF). Drug interactions were checked using AMH and BNF. The collected data were entered into the Microsoft Excel database and tested for anomalies to clean the data and improve their authenticity.

### 2.8. Data Analysis

Microsoft Excel 2016 was used to enter the collected data. Basic descriptive statistics and other advanced statistical tests were performed via the Statistical Process for Social Sciences 26 (IBM^®^ SPSS^®^). The normality of total BMQ scores and scores of all three screens were checked via the Kolmogorov–Smirnov and Shapiro–Wilk tests. Simple descriptive statistics were obtained for total BMQ scores and MAI scores. Associations between total BMQ score and sociodemographic variables were obtained via the Mann–Whitney U test and the Kruskal–Wallis test.

### 2.9. Outcomes of the Study

There are three major outcomes of this study. These outcomes include the assessment of medication adherence using the BMQ, the evaluation of the influence of sociodemographic factors on medication adherence, and the assessment of medication appropriateness using the MAI in heart failure patients attending the cardiovascular clinic at the NHK, Sri Lanka.

## 3. Results

A total of 325 patients were recruited. Most of them were male (74.2%, n = 241) and older, with a mean age of 64.6 ± 9.9 years (Table 1). Also, a summary of the study results were presented in Figure 1.

### 3.1. Baseline Medication Adherence Assessment

The mean total BMQ score for the sample (n = 325) was 1.16 ± 0.10, with scores ranging from 7.56 to 0.00. The mean scores for regimen, belief, and recall screens were 0.74 ± 0.68, 0.12 ± 0.30, and 0.32 ± 0.43, respectively. The distribution of total BMQ scores, along with the scores for the regimen, belief, and recall screens, was illustrated using boxplots in Figure 2.

Most of the study participants (52.0%) were classed as probable poor adherent to their medications, whereas only 11.7% of participants were adherent. Variable levels of medication adherence among the study participants (n = 325) were shown in Figure 3. 

At 95% confidence interval, statistically significant associations were found between medication adherence and age (*p* = 0.039), gender (*p* = 0.003), and education level (*p* = 0.027). No associations were reported between medication adherence and marital status (*p* = 0.297), living arrangement (*p* = 0.510), and employment status (*p* = 0.568).

The mean BMQ score was 1.62 in the age category below 45 years old and the age category of 60–74 years had a mean score of 1.09. This reveals relatively greater adherence among 60–74-year-old patients than among the other groups. Furthermore, the mean total BMQ score of females (1.34) was greater than that of males (1.10), suggesting a lower level of adherence among female patients. Additionally, patients belonging to the no schooling category had the highest mean total BMQ score, which was 1.87, suggesting a lower level of adherence than the other educational groups (Table 2).

### 3.2. Outcomes of the Medication Appropriateness Assessment

The mean MAI score for all the study participants (n = 325) was 0.56 ± 0.25, and the median MAI score was 0.55 (0.71–0.40). A boxplot illustration of the MAI scores (n = 325) was presented in Figure 4.

Basic descriptive statistics of total BMQ scores and MAI scores for total sample (n = 325) were presented in Table 3.

## 4. Discussion

Heart failure is a highly prevalent chronic disease that requires life-long treatment to slow disease progression and improve quality of life. In the current study, only approximately one-third of the study participants reported acceptable medication adherence, while most of the participants reported a lower level of medication adherence. This finding aligns with a study conducted in Sri Lanka among acute coronary syndrome patients. Most of the patients had low and probable low adherence rather than acceptable adherence to prescribed medications [31]. However, this finding is inconsistent with a study conducted among heart failure patients attending cardiac clinics in the north of Jordan, which reported 33.5% high adherence and 19.5% good adherence levels [16]. Additionally, another study conducted among heart failure patients admitted to the cardiology department in the provincial integrated hospital in Poland reported higher adherence and moderate adherence (70%), with a low adherence proportion of 26.7% [36]. Therefore, the deviations in the present findings from those studies could be due to variations in treatment settings, patient knowledge, and other setting-specific factors. The poor adherence to medications among the participants in this study may have been caused by various factors. Patients did not have a proper understanding of their medications, including the names of the prescribed medications, the reasons for prescribing them, and the importance of taking the prescribed medications properly.

In this study, the association between medication adherence and patient age was similar to the findings of another cross-sectional study conducted among heart failure patients in Jordan [16]. However, several other studies conducted among heart failure patients admitted to hospitals have reported that age does not have a significant effect on adherence [36,37]. Most of the study participants in this study (N = 195, 60.0%) represented the age category of 60–74 years old (Table 1), and this association can be directly linked to the majority of elderly participants. Similar to the findings of this study, another study reported an association between medication adherence and gender [38]. They have further shown that females have a lower adherence level than males do. This aligns with the findings of this study, which used the mean total BMQ score of males and females to determine the status of adherence. However, these findings contrast with those of a study conducted in Sri Lanka among a cardiovascular patient cohort, indicating that gender did not have a notable influence on medication adherence [31].

Previous studies have revealed that the education level of patients is associated with medication adherence [16,36]. This finding is in accordance with the findings of this study. Differences in memory and understanding of the disease and the medicines used may have contributed to this association. According to the mean total BMQ scores observed for each education level in this study, the adherence level was relatively lower in patients who had no schooling than in those with other education levels. This finding confirms the importance of patient knowledge of medications for improved medication adherence, as patients with no school education are highly unlikely to ascertain medication knowledge. This finding is confirmed by a study performed among heart failure patients, indicating education level that was higher than high school was associated increased level of medication adherence [16]. Additionally, in Sri Lanka, poor health literacy can be observed even among educated people, and this can be more significant in patients with no schooling. Furthermore, when the education level is improved, patients may seek information about their medications, increasing their access to healthcare resources more than others do. These findings may be attributed to differences in medication adherence across different education levels in this study. Although no statistically significant associations were observed between total BMQ scores and marital status, living arrangement, or employment status in this study, certain previous findings are inconsistent with those findings [16,36]. These variations are apparent, as the differences in these variables could be related to variations in the social and cultural impacts of different study settings.

Globally, medication non-adherence among heart failure patients is considerably high [27]. This is consistent with the findings of our study, which also indicated suboptimal medication adherence among heart failure patients. Poor adherence among these patients may lead to serious consequences such as increased mortality rates, frequent emergency department visits, worse cardiac events, hospital readmissions, and decreased quality of life of patients [27,28,39]. Therefore, these implications highlight the critical need of targeted interventions to improve medication adherence among heart failure patients by the multidisciplinary healthcare team managing heart failure patients.

Theoretically, the MAI score should be zero for a medication list of a patient when all the medicines prescribed are completely appropriate. In this study, the mean MAI score was 0.56 ± 0.25, which contrasts with the results of a study conducted in Kuwait that used the MAI to determine the prevalence of PIM [40]. They have shown that the mean MAI score per patient was 5.8 ± 5.8, revealing a considerable difference compared to this study. Patients attending the clinic are frequently subjected to relevant diagnostic methods such as ECHO, electrocardiogram (ECG), and other laboratory tests to facilitate the correct diagnosis and determine the current condition of patients to update their medications. Additionally, standard treatment guidelines are used by doctors for prescribing medications in the study setting, leading to prescribing a standard set of medications for heart failure. These findings may explain the improved appropriateness of the prescribed medications in this study.

In this study, an assessment of medication adherence and medication appropriateness was performed with validated study tools that have been used frequently both locally and internationally. The BMQ is an effective method for assessing medication adherence in patients with chronic illness with high specificity, accuracy, and sensitivity. It has been translated and validated in Sri Lankan settings, leading to more accurate results [30]. Also, a successful accomplishment of reaching the targeted population in the cardiac clinic was achieved, ensuring the reliability of the study findings. This study has a few limitations. Mainly, this study was conducted in a single healthcare institution as there was a limited availability of time and resources for the study; this may limit the generalisability of findings to other healthcare settings with different patient population and healthcare practices. This is a cross-sectional study design, and it may not capture the assessment of changes in medication adherence over time. Also, convenience sampling method was used to recruit study participants, and this may cause selection bias in the study. The use of the BMQ as an interviewer-administered tool in this study represents a deviation from its original design as a self-administered instrument and should be considered a methodological limitation. However, this approach was adopted to mitigate the risk of incomplete or inaccurate responses due to limited health literacy among patients in the Sri Lankan context. The use of only one tool with self-reporting to assess medication adherence is also one of the limitations because it may not capture the complexity of medication adherence behaviours completely. Thus, medication adherence was measured indirectly and, it may influence the findings by recall bias. Therefore, a combination of both direct and indirect methods is recommended to improve the precision and reliability of adherence assessment, as the exclusive use of an indirect method may compromise the precision of adherence estimates. Moreover, potential interrelationships among independent variables such as age, gender, and education level were not evaluated in this study, and future studies can explore these associations to understand the contribution of independent variables on medication adherence to introduce interventions to improve patient’s medication. In addition, collaboration with the treating team and access to updated guidelines are needed for a more accurate assessment of MAI. Thus, the achieved sample size was 325 because referring to each clinic book and ECHO report was necessary to recruit patients for the study, which was a time-consuming process. This results in a small reduction in the overall sample size with a slight increase in margin of error and decrease in statistical power compared to the intended sample size.

## 5. Conclusions

These findings indicate that most heart failure patients have poor medication adherence, and that the level of medication adherence is significantly impacted by their age, gender, and education level. The appropriateness of medications prescribed for heart failure patients at the cardiology clinic can be considered acceptable according to the MAI screen. As future measures, it is important to conduct both quantitative and qualitative studies to further examine the barriers to medication adherence, the factors influencing medication adherence, and patients’ perspectives regarding medication adherence among heart failure patients. Also, future research should focus on designing and evaluating structured interventions aimed at improving medication adherence among patients with heart failure. Conducting large-scale, multi-centre randomised controlled trials would generate high-quality evidence on effective strategies for enhancing adherence in this population.

## Figures and Tables

**Figure 1 pharmacy-13-00101-f001:**
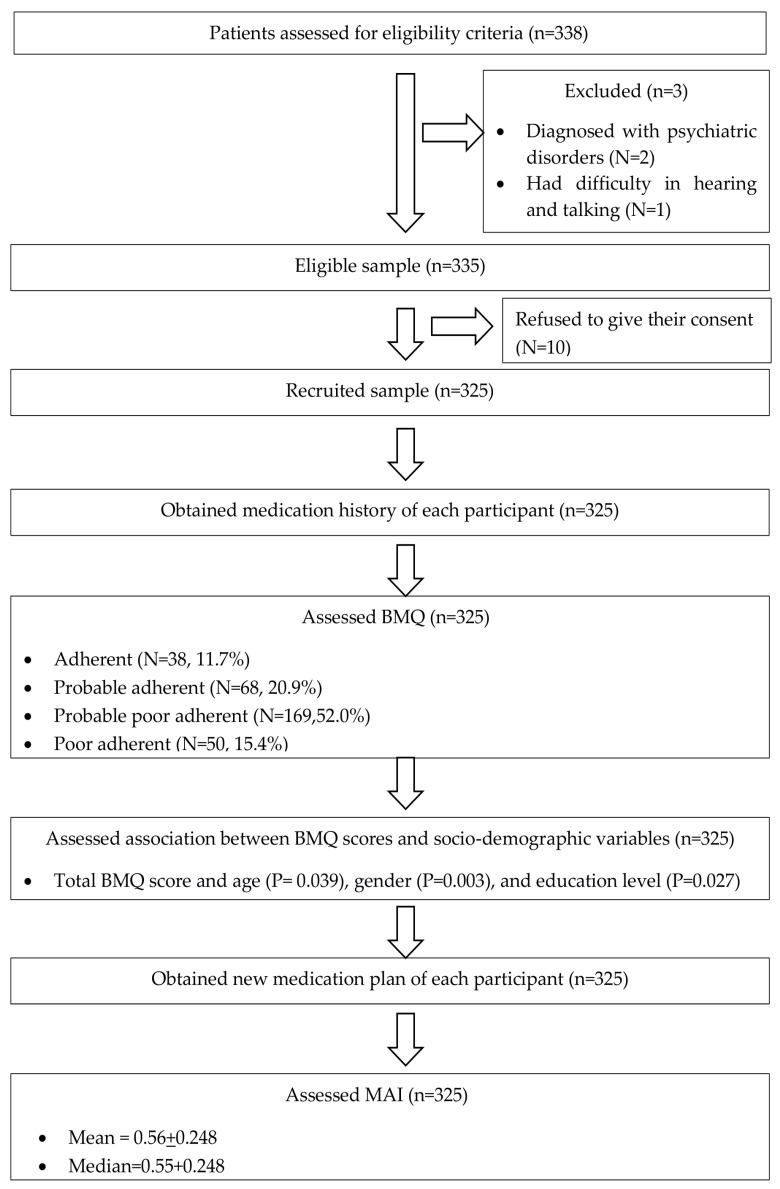
Summary of the study results.

**Figure 2 pharmacy-13-00101-f002:**
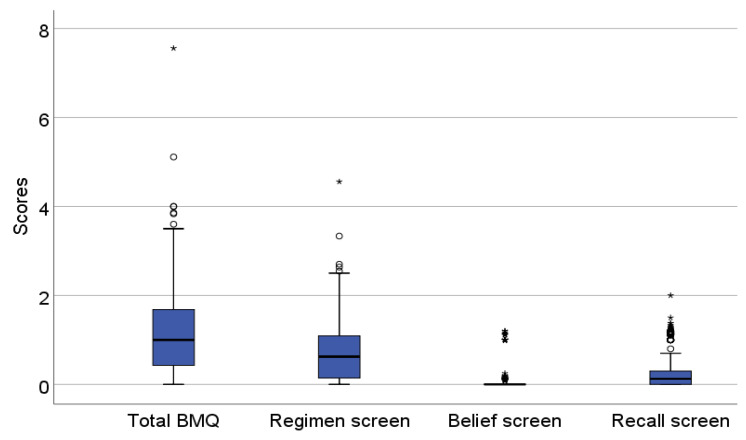
Boxplot illustration of distribution for scores of total BMQ and 3 screens.

**Figure 3 pharmacy-13-00101-f003:**
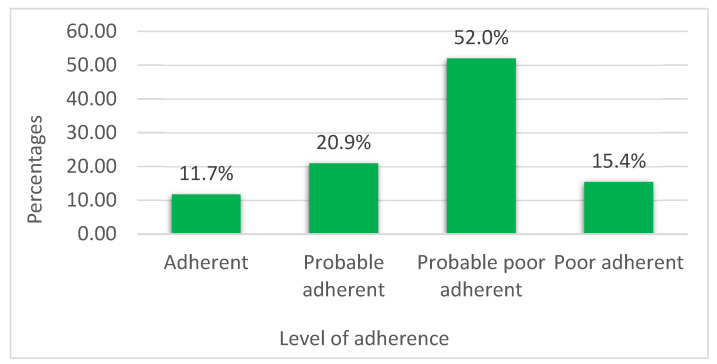
Variable levels of medication adherence among the participants.

**Figure 4 pharmacy-13-00101-f004:**
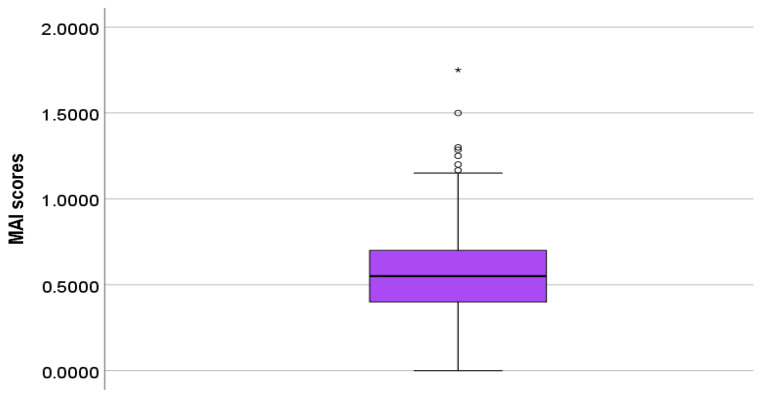
Boxplot illustration of MAI scores (n = 325).

**Table 1 pharmacy-13-00101-t001:** Characteristics of the study participants (n = 325).

Characteristics	Category	Frequency (N) and Percentages (%)
Age (years)	Below 45	10 (3.1)
	45–59	80 (24.6)
	60–74	195 (60.0)
	75 or above	40 (12.3)
Gender	Male	241 (74.2)
	Female	84 (25.8)
Marital status	Unmarried	13 (4.0)
	Married	272 (83.7)
	Widowed/divorced	40 (12.3)
Education level	No schooling	10 (3.1)
	Grades 1 to 11	251 77.2)
	Up to A/L, graduated or others	64 (19.7)
Current	Yes	87 (26.8)
employment status	No	238 (73.2)
Living arrangement	Alone	18 (5.5)
	With spouse	69 (21.2)
	With spouse and children	180 (55.4)
	Others	58 (17.9)

**Table 2 pharmacy-13-00101-t002:** Mean total BMQ scores of categories.

Characteristics	Category	Mean Total BMQ Score
Age (years)	Below 45	1.62
	45–59	1.13
	60–74	1.09
	75 or above	1.50
Gender	Male	1.10
	Female	1.34
Education level	No schooling	1.87
	Grades 1 to 11	1.17
	Up to A/L, graduated or others	1.04

**Table 3 pharmacy-13-00101-t003:** Basic descriptive statistics of total BMQ scores and MAI scores.

	Total BMQ Score (n = 325)	MAI Score (n = 325)
Mean	1.16	0.56
Median	1.00	0.55
Quartile (Q1)	0.43	0.40
Quartile (Q2)	1.00	0.55
Quartile (Q3)	1.70	0.71
Interquartile range (IQR)	1.27	0.31

## Data Availability

The data presented in this study are available on request from the corresponding author.

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
