# Peer review of "Evaluation of Medication Adherence and Appropriateness Among Heart Failure Patients Attending the Cardiac Clinic at a Tertiary Care Hospital: A Cross-Sectional Observational Study"

_pharmacy, 2025, doi:10.3390/pharmacy13040101_

Round 1
Reviewer 1 Report
Comments and Suggestions for Authors
The study " Evaluation of Medication Adherence and Appropriateness among Heart Failure Patients Attending the Cardiac Clinic at a Tertiary Care Hospital: A Pharmacist-Led Cross-Sectional Study" aimed "to assess medication adherence, evaluate the influence of sociodemographic factors on medication adherence and assess medication appropriateness by a pharmacist for patients attending the cardiac clinic at National Hospital Kandy (NHK)."
Introduction provide enough background and is well written.
Material and Methods are appropriate and well presented.
The Results are adequately and clearly presented. Minor corrections needed.
Page 7, line 200 - Figure legend for Figure 3 is missing. More precisely, under Figure 3 the legend for Figure 2 was copied and pasted. Provide adequate figure legend.
Page 8, line 216 – check and correct the Figure numbering
The Discussion is well conducted but could be enriched. In order to enrich this section please present in more details the findings of Awad and Hanna (2019) study (page 10, line 271, reference number 33 in your manuscript).
How the obtained results may affect the role of a pharmacist, in general and in your country?
How can be the obtained results extrapolated to other clinics in the country? Consider mentioning this single-institution study design as a limitation of the study in one or two sentences.
Conclusions are based on the results.
Author Response
1. Summary |
|
|
Thank you very much for taking the time to review this manuscript. Please find the detailed responses below and the corresponding revisions/corrections highlighted in the re-submitted files.
|
||
3. Point-by-point response to Comments and Suggestions for Authors |
||
Comments 1: Page 7, line 200. Figure legend for figure 3 is missing. More precisely, under Figure 3 the legend for Figure 2 was copied and pasted. Provide adequate figure legend
|
||
Response 1: Thank you for pointing this out. The legend under Figure 3 was mistakenly copied from Figure 2 and, this has now been corrected in the revised manuscript. (Page 8, line 212)
|
||
Comments 2: Page 8, line 216. Check and correct the figure numbering
|
||
Response 2: Thank you for your observation. Figure numbering was corrected in revised manuscript. (Page 9, line 230)
Comments 3: Please present in more details the findings of Award and Hanna (2019) study (page 10, line 271, reference number 33 in your manuscript)
Response 3: Thank you for your valuable suggestion. We have revised the relevant section to elaborate further on the findings. According to the study of Award and Hanna (2019), the mean (SD) MAI score per patient was 5.8(5.8). But, the mean MAI score per patient of our study is 0.56(0.25). So, there is a considerable difference in obtained MAI scores of these two studies. This was also mentioned in the revised manuscript in (Page 11, line 302-303)
Comments 4: How the obtained results may affect the role of a pharmacist, in general and in your country?
Response 4: Thank you for this insightful comment. Obtained results highlighted the crucial role of a pharmacist in identifying patient medication adherence and appropriateness of their medications. Generally, this showcases the importance of involvement of pharmacists in improving patient education, medication review and follow up of patients to improve the quality of life of patients. In our country, it can be mentioned that pharmacists are underutilized in clinical setting. These findings emphasize the importance of pharmacists in clinical settings, expanding their role in multidisciplinary health care teams. Overall, pharmacist extending into clinical roles will lead to improved patient care in managing chronic diseases efficiently.
Comments 5: How can be the obtained results extrapolated to other clinics in the country?
Response 5: We appreciate your question regarding this matter. Usually, medication non-adherence and polypharmacy issues are common in many clinics in the country. Also, the demographic and clinical characteristics of the study population are reflective of many other clinics in our country. Because most heart failure patients are suffering from other comorbidities resembling the other chronic disease patient clinics. Although multi-institutions study design may be needed for further precise generalisable conclusions, obtained findings can suggest useful insights to extrapolated to other clinics as well.
Comments 6: Consider mentioning this single-institution study design as a limitation of the study in one or two sentences.
Response 6: We appreciate your thoughtful suggestion regarding the study design. So, it was considered as a limitation and included in the revised manuscript. (Page 12, line 317-319)
|

Reviewer 2 Report
Comments and Suggestions for Authors
Dear authors,
I think your topic is critical and well described. Here are some of the considerations:
1. Line 26: The phrase "probable poor adherent level" is not standard terminology. Could you please clarify or use validated adherence categories?
2. Line 23 -27: The abstract includes numerous numerical values but does not provide 95% confidence intervals or effect sizes.
3. The introduction contains repetitive statistical data regarding CVD mortality and heart failure prevalence. Could the authors rewrite this section to avoid redundancy?
4. While the rationale is generally clear, the objective could be restated as a research question or hypothesis at the end of the introduction.
5. Line 127: The calculated sample size was 385, but the study was completed with 325 participants. The text states a “small reduction.” Explain whether this reduction affected statistical power.
6. Since convenience sampling was used, how do the authors address potential selection bias and its impact on generalizability?
7. Figure 2 is referenced twice with different contents (Boxplot and BMQ levels). Clarify the figures.
8. The results section lacks confidence intervals and p-values for most of the reported associations. To increase transparency, include these.
9. The mean MAI score is presented as "acceptable" without referencing validated thresholds. Why do you use MAI and not other tools?
10. The limitations section is relatively brief. Can the authors expand on how the use of one site, the lack of longitudinal data, and possible measurement bias might limit conclusions?
11. The conclusions suggest integrating pharmacists into heart failure teams. Could the authors expand on feasible strategies for doing so in Sri Lanka?
Author Response
1. Summary |
|
|
Thank you very much for taking the time to review this manuscript. Please find the detailed responses below and the corresponding revisions/corrections highlighted in the re-submitted files.
|
||
3. Point-by-point response to Comments and Suggestions for Authors |
||
Comments 1: Line 26: The phrase "probable poor adherent level" is not standard terminology. Could you please clarify or use validated adherence categories?
|
||
Response 1: Thank you for your valuable feedback. BMQ consists of 3 screens as regimen, belief, and recall screen. By referring to a previous similar study conducted using BMQ, patients were classified into four adherence levels based on the scores they obtained for each screen in BMQ. When no positive scores were reported across any of the screens, the patient was classified as adherent. A single positive score across the screens indicated probable adherence. Patients with positive scores on two screens were classified as probable poor adherents, while positive scores on all three screens were categorized as poor adherents.[1]
|
||
Comments 2: Line 23 -27: The abstract includes numerous numerical values but does not provide 95% confidence intervals or effect sizes.
|
||
Response 2: Thanks for highlighting this. A sentence regarding this was added to the abstract. (Page 1, line 21-22)
Comments 3: The introduction contains repetitive statistical data regarding CVD mortality and heart failure prevalence. Could the authors rewrite this section to avoid redundancy?
Response 3: Thank you for your valuable suggestion. We have carefully revised the introduction and removed a sentence on cardiovascular statistics to avoid redundancy while retaining the essential background information.
Comments 4: While the rationale is generally clear, the objective could be restated as a research question or hypothesis at the end of the introduction.
Response 4: Agree with your comment. The objective section in the introduction is modified accordingly. (Page 3, line 101-105)
Comments 5: Line 127: The calculated sample size was 385, but the study was completed with 325 participants. The text states a “small reduction.” Explain whether this reduction affected statistical power.
Response 5: Thank you for your comment. The study was completed with 325 participants, with a reduction of approximately 15.6% from the intended sample. Although this has led to a minor decrease in statistical power, the achieved sample size of 325 remains robust and adequate for estimating proportions with acceptable precision. Specifically, the margin of error has increased approximately to 5.4%, which is still within an acceptable range for cross-sectional studies. Therefore, the study’s ability to detect meaningful associations and draw valid conclusions remains largely unaffected. Nonetheless, this reduction has been acknowledged as a limitation. (Page 12, line 337-338)
Comments 6: Since convenience sampling was used, how do the authors address potential selection bias and its impact on generalizability?
Response 6: We acknowledge that selection bias may be caused by convenience sampling, as study participants were selected based on their willingness to participate and accessibility. Therefore, it was considered a limitation of this study, and we have rewritten it in the discussion section (Page 12, line 320-322), so that future studies should focus on random sampling and be conducted in multiple centers to improve the generalizability.
Comments 7: Figure 2 is referenced twice with different contents (Boxplot and BMQ levels). Clarify the figures.
Response 7: Thanks a lot, highlighting this issue. Corrected in the revised manuscript. (page 8 , line 212)
Comments 8: The results section lacks confidence intervals and p-values for most of the reported associations. To increase transparency, include these.
Response 8: Thank you for your valuable suggestion. This was included in the revised manuscript in result section. (Page 8, line 213-216)
Comments 9: The mean MAI score is presented as "acceptable" without referencing validated thresholds. Why do you use MAI and not other tools?
Response 9: Thank you for this insightful comment. MAI is focused on ten important criterions: indication, effectiveness, dosage, direction, drug-drug interactions, drug-disease interactions, practical, duration, therapeutic duplication and cost compared to equal alternatives. Also, it is a reliable, freely available and validated tool used to measure potentially inappropriate prescribing in different clinical settings[2]. Moreover, previous studies conducted in our country have also used this tool, which supports its relevance and applicability in our healthcare setting [3,4]. Because of these reasons, MAI was used to assess medication appropriateness rather than using other tools. The Medication Appropriateness Index (MAI) does not have established thresholds for categorising appropriateness levels. Instead, it serves as an indicative tool, where a score of zero reflects complete appropriateness of medication use. As the MAI score increases, it reflects a decline in appropriateness, with higher scores indicating more concerns or inappropriate prescribing elements.
Comments 10: The limitations section is relatively brief. Can the authors expand on how the use of one site, the lack of longitudinal data, and possible measurement bias might limit conclusions?
Response 10: Thank you for your valuable feedback. This study was conducted in a single institution because of the time constraints and limited resources. Also, the cross-sectional nature of this study may not capture the assessment of medication adherence changes over time. Convenience sampling method was used because study participants were recruited with the accessibility and their willingness to participate. So, there might be selection bias during the study. Therefore, we accept that future studies should focus on multi-center study design and random sampling to avoid those limitations. Hence, limitation section has been expanded accordingly in the revised manuscript. (Page 12, line 317-323)
Comments 11: The conclusions suggest integrating pharmacists into heart failure teams. Could the authors expand on feasible strategies for doing so in Sri Lanka?
Response 11: Although clinical pharmacy practice is still a novel approach in Sri Lankan public hospitals, pharmacists can be integrated into multidisciplinary team step by step. Pharmacist can be allocated to review discharge medications lists before dispensing to detect and correct potential drug-related problems to optimize discharge medications lists and clinic medication lists as an initial step of integration. Also, pharmacist can engage in medication counseling at the time of dispensing during discharge and at clinics. The conclusion is modified according to this feedback. (Page 12, line 343-348) |

Reviewer 3 Report
Comments and Suggestions for Authors
General Comments:
The manuscript addresses a relevant and timely topic within the broader context of medication adherence. The authors are to be commended for their careful consideration of inclusion criteria and participant selection, which appear well-defined and appropriate. Despite some inaccuracies and points that require clarification (outlined below), the rationale for exploring beliefs about medicines is, in essence, pertinent.
However, the study presents several critical limitations, both in methodology and in data interpretation, that would require substantial revision to align the manuscript with publication standards. I outline below specific points for the authors' consideration.
- Specific Comments:
Relevance of Section 1.4
The relevance of point 1.4 is unclear. While it may be valid to argue in the Discussion section that low adherence could partially stem from limited pharmacist involvement in therapeutic adherence strategies, this argument is out of place in the Introduction. Moreover, the manuscript seems to repeatedly draw attention to the “role of the pharmacist,” which comes across as forced. The value of any healthcare professional is best reflected in the strength of the study itself; such merits tend to emerge organically through well-conducted research rather than through repeated assertions.
- Data Protection Measures
The current description of data security—password-only access—is concerningly weak. Could the authors clarify whether multiple authentication steps were in place for accessing the data stored on Google Cloud? Was data protection regulation, such as GDPR, duly considered? Were the access credentials institutional? How many team members had access to the raw data? These are key points that need to be clearly addressed to ensure data privacy and integrity.
- Use of the BMQ in Interview Format
Have other studies employed the BMQ score using an interview format instead of a self-administered questionnaire? If so, these precedents should be acknowledged. Furthermore, the potential drawbacks of this methodological change must be clearly discussed, including the possible impact on response reliability and the overall consistency of the instrument.
- Figure 1 – Quality Concerns
Figure 1 is of poor visual quality and should be replaced with a higher-resolution version to ensure clarity and professionalism in presentation.
- Line 205 – Redundancy
The statement “A higher BMQ score indicates a lower level of adherence” is redundant, as this information has already been stated earlier in the manuscript.
- Sociodemographic Data
The sociodemographic characterization of participants should be presented in the main body of the manuscript, not relegated to supplementary materials. Furthermore, statistical analyses should be conducted to assess whether relationships exist among variables (e.g., is there a higher proportion of males with higher education levels?). These associations could provide valuable insights.
- Brevity of Results
The results section is too brief. It is likely that more data were collected and could be explored. At the very least, beyond sociodemographic breakdowns, the authors should analyze individual item responses within the scales used.
- Indirect Measurement of Adherence
The instrument used measures adherence only indirectly. This limitation must be explicitly acknowledged by the authors.
- Self-Report Bias
Self-reporting is a known limitation in adherence studies and should be transparently stated in the limitations section.
- Lack of Context in Literature Citations (Lines 225–227 and 236–238)
The studies cited in these lines should be contextualized—at a minimum, include the country or setting in which each study was conducted to help readers understand the relevance and comparability of findings.
- Discussion Section – Insufficient Depth
The discussion is underdeveloped. It fails to address specific medications and the implications of their (non)adherence. More importantly, it does not delve into the real-world consequences of poor adherence or contextualize findings within existing literature. 12. A major revision is needed to strengthen both the interpretation of results and the presentation of limitations.
- Conclusion – Misalignment with Findings
The conclusion appears misaligned with the study’s actual findings. The emphasis on the pharmacist’s role in improving adherence is not supported by the data. No outcome in the study evaluates the impact of pharmacist-led interventions, nor is there evidence that pharmacist administration of the BMQ, rather than that by another qualified health professional, altered results. The BMQ is a multidisciplinary tool employed by a wide range of health researchers, not exclusively pharmacists.
I hope these comments assist both the editorial team and the authors in enhancing the quality and scientific rigor of the manuscript.
Author Response
1. Summary |
|
|
Thank you very much for taking the time to review this manuscript. Please find the detailed responses below and the corresponding revisions/corrections highlighted in the re-submitted files.
|
||
3. Point-by-point response to Comments and Suggestions for Authors |
||
Comments 1: Relevance of Section 1.4. The relevance of point 1.4 is unclear. While it may be valid to argue in the Discussion section that low adherence could partially stem from limited pharmacist involvement in therapeutic adherence strategies, this argument is out of place in the Introduction. Moreover, the manuscript seems to repeatedly draw attention to the “role of the pharmacist,” which comes across as forced. The value of any healthcare professional is best reflected in the strength of the study itself; such merits tend to emerge organically through well-conducted research rather than through repeated assertions.
|
||
Response 1: Thank you for your valuable feedback. BMQ consists of 3 screens as regimen, belief, and recall screen. By referring to a previous similar study conducted using BMQ, patients were classified into four adherence levels based on the scores they obtained for each screen in BMQ. When no positive scores were reported across any of the screens, the patient was classified as adherent. A single positive score across the screens indicated probable adherence. Patients with positive scores on two screens were classified as probable poor adherents, while positive scores on all three screens were categorized as poor adherents.[1]
|
||
Comments 2: Data Protection Measures The current description of data security—password-only access—is concerningly weak. Could the authors clarify whether multiple authentication steps were in place for accessing the data stored on Google Cloud? Was data protection regulation, such as GDPR, duly considered? Were the access credentials institutional? How many team members had access to the raw data? These are key points that need to be clearly addressed to ensure data privacy and integrity.
|
||
Response 2: I appreciate your important observations regarding data security and privacy. I confirm that all raw data were stored in a secure, university-managed Google Workspace (G-Suite) cloud storage system. Access to this storage was protected using two-factor authentication and linked to institutional email accounts, ensuring enhanced data security. Access to the raw data was strictly limited to the first and second authors, both of whom are affiliated with the hosting university and were directly involved in data handling and analysis. The access credentials were institutional, and access was governed by the university’s data governance and information security protocols. No personally identifiable information was shared beyond this team. The analysed and de-identified data were subsequently shared with the other co-authors solely for the purposes of interpretation and manuscript development. I also confirm that data protection principles consistent with GDPR and institutional ethical standards were considered throughout the study to ensure compliance with relevant data privacy regulations. The manuscript also edited adding missing information. (Page 3, line l16-124)
Comments 3: Use of the BMQ in Interview Format Have other studies employed the BMQ score using an interview format instead of a self-administered questionnaire? If so, these precedents should be acknowledged. Furthermore, the potential drawbacks of this methodological change must be clearly discussed, including the possible impact on response reliability and the overall consistency of the instrument.
Response 3: Thank you for your insightful comment. BMQ was used as an interviewer administered tool because of limited healthcare literacy among many patients in our country. Many of them are unable to recall or pronounce drug names either by generic name or brand name. So, it was not used as a self-administered tool to prevent the collection of possible inaccurate and incorrect information. But, we used the patients’ clinic books as a reference during the interview process to minimise recall bias and improve accuracy. This helped to ensure reliability and consistency of collected information. It was mentioned in the revised manuscript as well (Page 4, line 148-152). Also, similar studies have been conducted in our country using the BMQ as an interviewer- administered tool[3]. This was also mentioned in the revised manuscript in line (Page 4, line 148 -152). This is also discussed as a limitation under the discussion. (Page12, line 320 -324)
Comments 4: Figure 1 – Quality Concerns Figure 1 is of poor visual quality and should be replaced with a higher-resolution version to ensure clarity and professionalism in presentation.
Response 4: Thank you for your observation. Figure 1 was replaced with a higher-resolution version. (Page 7)
Comments 5: Line 205 – Redundancy The statement “A higher BMQ score indicates a lower level of adherence” is redundant, as this information has already been stated earlier in the manuscript. Response 5: We agree with your comment. So, the redundant sentence has been removed from the manuscript to prevent the repetition.
Comments 6: Sociodemographic Data The sociodemographic characterization of participants should be presented in the main body of the manuscript, not relegated to supplementary materials. Furthermore, statistical analyses should be conducted to assess whether relationships exist among variables (e.g., is there a higher proportion of males with higher education levels?). These associations could provide valuable insights.
Response 6: I appreciate your concern in this matter, and we have now moved the socio-demographic characteristics of study participants table to the main body of the manuscript (Page 5, line 201) Our main objective was to observe whether there are associations between socio-demographic factors and medication adherence. Assessing relationships among variables was beyond the objectives of our study. Therefore, we did not perform the subgroup analysis, but we accept that future studies should further explore these associations to provide a better understanding of medication adherence and related factors.
Comments 7: Brevity of Results The results section is too brief. It is likely that more data were collected and could be explored. At the very least, beyond sociodemographic breakdowns, the authors should analyze individual item responses within the scales used. Response 7: We appreciate your thoughtful comment. This study was conducted within a limited three-month duration. Within this timeframe, all collected data were thoroughly analysed and presented in the results section. We confirm that there are no additional data beyond what has already been reported.
Comments 8: Indirect Measurement of Adherence The instrument used measures adherence only indirectly. This limitation must be explicitly acknowledged by the authors. Response 8: Thank you for your valuable comment. We acknowledge that the BMQ used in this study primarily measures adherence indirectly. It relies on patient-reporting rather than objective measures. The use of both direct and indirect methods is expected to enhance the accuracy and reliability of medication adherence assessment. So, there may be issues with the precision of adherence estimates due to the exclusive use of an indirect method. Therefore, this limitation has been addressed in the revised manuscript (Page 12, line 326-330)
Comments 9: Self-Report Bias Self-reporting is a known limitation in adherence studies and should be transparently stated in the limitations section.
Response 9: Thank you for your suggestion. We acknowledge that self-reporting may lead to recall bias and indirect adherence assessment methods associate with some issues causing over or underestimation of adherence compared to direct methods. Therefore, this was mentioned in the revised manuscript in limitation section. (Page 12, line 324-330)
Comments 10: Lack of Context in Literature Citations (Lines 225–227 and 236–238) The studies cited in these lines should be contextualized—at a minimum, include the country or setting in which each study was conducted to help readers understand the relevance and comparability of findings.
Response 10: Thank you for your valuable feedback. We have revised the cited sections including the country or settings where the study was conducted to include contextual details. (Page 10, line 238, 240-241, 251-252)
Comments 11: Discussion Section – Insufficient Depth The discussion is underdeveloped. It fails to address specific medications and the implications of their (non)adherence. More importantly, it does not delve into the real-world consequences of poor adherence or contextualize findings within existing literature.
Response 11: Thank you for your valuable suggestion. According to the objectives of our study, we did not assess adherence issues specific to individual medications. We focused on overall adherence patterns rather than drug-specific nonadherence in discussion section. However, we have included real-world consequences of poor adherence in the discussion to contextualize the findings. (Page 11, line 282-287). Moreover, we agree that exploring medication specific nonadherence will provide valuable insights and suggest this as an area for future studies.
Comments 12: A major revision is needed to strengthen both the interpretation of results and the presentation of limitations.
Response 12: Thank you for your comment. As mentioned earlier, all the collected data have been thoroughly analysed and findings were presented according to the predefined objectives. No additional results are available beyond what has already been included. However, limitation section has been expanded accordingly in the revised manuscript. (Page 12, line 314 - 335)
Comments 13: Conclusion – Misalignment with Findings The conclusion appears misaligned with the study’s actual findings. The emphasis on the pharmacist’s role in improving adherence is not supported by the data. No outcome in the study evaluates the impact of pharmacist-led interventions, nor is there evidence that pharmacist administration of the BMQ, rather than that by another qualified health professional, altered results. The BMQ is a multidisciplinary tool employed by a wide range of health researchers, not exclusively pharmacists.
Response 13: We agree with your comment. So, the conclusion has been modified according to your and other reviewers’ comments. (Page 12, line 340-345) |

Round 2
Reviewer 3 Report
Comments and Suggestions for Authors
The authors have attempted to address most of the comments. However, those comments that required deeper critical reflection and more substantial analytical work were not satisfactorily addressed. I maintain my reservations, particularly regarding the following points:
The response to my Comment 1 from the previous review does not appear to fully correspond to the issues I raised. Additionally, in the same subsection, a motivation for conducting the study has now been added—yet the research does not address it. Specifically, the study presents no results that demonstrate the pharmacist’s ability to identify higher or lower adherence to therapy. To achieve such a goal, multiple pharmacists would need to be assessed, and there would need to be a reliable benchmark for actual adherence levels within a given population, which could then be compared with the pharmacists’ assessments. The sentence added to the conclusion on this matter is similarly inappropriate.
In my previous Comment 6, I emphasized the importance of evaluating the independence of variables and potential associations among those initially classified as "independent." The authors considered this evaluation to be nonessential and did not even include it as a limitation. I stand by my original opinion and believe this should be reconsidered, as the current approach may lead to premature or unsupported conclusions.
Regarding the conclusion, I had pointed out the inadequacy of the emphasis placed on the role of the pharmacist. Not only was this emphasis maintained, but it was further reinforced. This is not justified, as the results do not evaluate the pharmacist’s ability or potential to assess therapeutic adherence.
Author Response
Comments 1: The response to my Comment 1 from the previous review does not appear to fully correspond to the issues I raised. Additionally, in the same subsection, a motivation for conducting the study has now been added—yet the research does not address it. Specifically, the study presents no results that demonstrate the pharmacist’s ability to identify higher or lower adherence to therapy. To achieve such a goal, multiple pharmacists would need to be assessed, and there would need to be a reliable benchmark for actual adherence levels within a given population, which could then be compared with the pharmacists’ assessments. The sentence added to the conclusion on this matter is similarly inappropriate. |
Response 1: Thank you for your helpful clarification. We now understand that the previously added rationale, which emphasized the pharmacist’s ability to assess adherence, was not appropriately aligned with the scope and findings of our study. Accordingly, section 1.4 has been revised to maintain the consistency. (Page 3 , line 90-108)
|
Comments 2: In my previous Comment 6, I emphasized the importance of evaluating the independence of variables and potential associations among those initially classified as "independent." The authors considered this evaluation to be nonessential and did not even include it as a limitation. I stand by my original opinion and believe this should be reconsidered, as the current approach may lead to premature or unsupported conclusions. |
Response 2: Thank you for highlighting the importance of evaluating the independence of variables and potential associations among them. We accept that interrelationship among the independent variables may influence the associations between medication adherence and sociodemographic factors. However, in our study, our primary focus on association between medication adherence and individual sociodemographic factors. So, as suggested, this was concerned as a limitation, and revised in the discussion section. (page 12, line 336-339). Also, we hope to address this area more comprehensively in future studies. Comments 3: Regarding the conclusion, I had pointed out the inadequacy of the emphasis placed on the role of the pharmacist. No t only was this emphasis maintained, but it was further reinforced. This is not justified, as the results do not evaluate the pharmacist’s ability or potential to assess therapeutic adherence. Response 3: Thank you for your feedback. In our country, the incorporation of a pharmacist in assessing medication adherence and appropriateness is particularly important, as pharmacists are often underutilised in the clinical setting and are not typically involved in these activities. In this study, the pharmacist successfully assessed the medication adherence levels and evaluated the appropriateness of prescribed medications among heart failure patients attending the cardiac clinic. Based on your feedback and that of other reviewers, we have revised the conclusion to ensure it accurately reflects the study’s findings. (page 12, line 350-353, line 357-360)
|

Round 3
Reviewer 3 Report
Comments and Suggestions for Authors
I appreciate the revisions made by the authors, who have addressed some of the suggestions and observations I raised in previous rounds. Certain aspects of the manuscript have indeed improved in clarity and structure.
However, the major methodological limitations identified earlier remain largely unresolved. In particular, there is a persistent issue that has been present across all versions of the manuscript: the authors continue to refer to the "success" of the pharmacist-led intervention or evaluation.
While I understand the authors’ interest in highlighting this point—and I acknowledge that such an outcome may have been a motivating factor for conducting the study—the current study design does not provide sufficient evidence to support that claim. This concern has already been noted in previous comments, yet the authors neither revised the manuscript accordingly nor provided any justification or response addressing the issue.
For example, the sentence in lines 350–351: “Also, the pharmacist was successfully assessed the medication adherence” remains unchanged. This is problematic, as the statement implies an outcome (success) that is not adequately supported by the methodological rigor or design of the study.
I encourage the authors to revisit this aspect carefully, either by removing or significantly rephrasing such claims
Author Response
Comments 1: I appreciate the revisions made by the authors, who have addressed some of the suggestions and observations I raised in previous rounds. Certain aspects of the manuscript have indeed improved in clarity and structure. However, the major methodological limitations identified earlier remain largely unresolved. In particular, there is a persistent issue that has been present across all versions of the manuscript: the authors continue to refer to the "success" of the pharmacist-led intervention or evaluation. While I understand the authors’ interest in highlighting this point—and I acknowledge that such an outcome may have been a motivating factor for conducting the study—the current study design does not provide sufficient evidence to support that claim. This concern has already been noted in previous comments, yet the authors neither revised the manuscript accordingly nor provided any justification or response addressing the issue. For example, the sentence in lines 350–351: “Also, the pharmacist was successfully assessed the medication adherence” remains unchanged. This is problematic, as the statement implies an outcome (success) that is not adequately supported by the methodological rigor or design of the study. I encourage the authors to revisit this aspect carefully, either by removing or significantly rephrasing such claims
|
Response 1: Thank you for your continuous feedback to improve the manuscript. We agree with your clarification regarding the current study design not supporting a few of the claims mentioned in the article. We accept that randomized controlled interventional studies in multiple locations with the interventions carried out by pharmacist is essential for such claims. Therefore, we have carefully reviewed the manuscript and revised such claims throughout matching the current study design. Section 1.4 is modified (page 3, line 95 – 98). Such claims are removed from the discussion (page 11, line 288 – 290). The sentence you mentioned in the lines 350 – 351 has been removed and the conclusion has been modified accordingly (page 12, line 343 -347). Also, please note that we have revised the study title removing the “pharmacist-led” phrase to match with the study design (page 1, line 4)
|
